# N-Palmitoyl-D-Glucosamine Inhibits TLR-4/NLRP3 and Improves DNBS-Induced Colon Inflammation through a PPAR-α-Dependent Mechanism

**DOI:** 10.3390/biom12081163

**Published:** 2022-08-22

**Authors:** Irene Palenca, Luisa Seguella, Alessandro Del Re, Silvia Basili Franzin, Chiara Corpetti, Marcella Pesce, Sara Rurgo, Luca Steardo, Giovanni Sarnelli, Giuseppe Esposito

**Affiliations:** 1Department of Physiology and Pharmacology “V. Erspamer”, Sapienza University of Rome, Piazzale Aldo Moro 5, 00185 Rome, Italy; 2Department of Clinical Medicine and Surgery, University of Naples “Federico II”, 80131 Naples, Italy; 3Department of Psychiatry, Giustino Fortunato University, 82100 Benevento, Italy

**Keywords:** micronized N-palmitoyl-D-glucosamine, ulcerative colitis, IBD, intestinal inflammation, toll-like receptors, PPARs, NLRP3, intestinal barrier

## Abstract

Similar to canine inflammatory enteropathy, inflammatory bowel disease (IBD) is a chronic idiopathic condition characterized by remission periods and recurrent flares in which diarrhea, visceral pain, rectal bleeding/bloody stools, and weight loss are the main clinical symptoms. Intestinal barrier function alterations often persist in the remission phase of the disease without ongoing inflammatory processes. However, current therapies include mainly anti-inflammatory compounds that fail to promote functional symptoms-free disease remission, urging new drug discoveries to handle patients during this step of the disease. ALIAmides (ALIA, autacoid local injury antagonism) are bioactive fatty acid amides that recently gained attention because of their involvement in the control of inflammatory response, prompting the use of these molecules as plausible therapeutic strategies in the treatment of several chronic inflammatory conditions. N-palmitoyl-D-glucosamine (PGA), an under-researched ALIAmide, resulted in being safe and effective in preclinical models of inflammation and pain, suggesting its potential engagement in the treatment of IBD. In our study, we demonstrated that micronized PGA significantly and dose-dependently reduces colitis severity, improves intestinal mucosa integrity by increasing the tight junction proteins expression, and downregulates the TLR-4/NLRP3/iNOS pathway via PPAR-α receptors signaling in DNBS-treated mice. The possibility of clinically exploiting micronized PGA as support for the treatment and prevention of inflammation-related changes in IBD patients would represent an innovative, effective, and safe strategy.

## 1. Introduction

Inflammatory Bowel Disease (IBD) is a complex of chronic and relapsing diseases of the gastrointestinal (GI) tract that converges environmental, microbial, immunological, and genetic factors both in humans [1] and dogs [2]. In humans, these heterogeneous GI disorders present as two major clinical phenotypes, ulcerative colitis (UC) and Crohn’s disease (CD), which are characterized by periods of remission and flare-ups of the disease, in which diarrhea, visceral hypersensitivity, and fever are commonly referred [3]. Acute inflammation markedly impairs intestinal physiology and function, and persistent alterations are often observed after the resolution of intestinal inflammation, consistently suggesting a role for inflammatory effects in generating the symptoms that occur during remission in patients with IBD. These long-term changes involve motility, abnormal secretion, and altered visceral sensation, and no effective treatments are currently available to handle this phase of the disease. In recent decades, IBD has alarming emerged in Western countries, predominantly UC, suggesting that this epidemiological evolution is likely related to the westernization of lifestyle associated with changes in diet, antibiotic use, hygiene status, and microbial exposures [4]. UC primarily involves confluent inflammation of the colonic mucosa and impairs the epithelial barrier integrity, and intestinal homeostasis. The consequent abnormal translocation of luminal microbes and their products across the impaired intestinal barrier leads to robust activation of resident macrophages and antigen-presenting cells (APCs), and the massive release of Tumor Necrosis Factor- α (TNF-α), Interleukin-1β (IL-1β), IL-13, IL-9, IL-23, IL-36, and other pro-inflammatory mediators [5]. This acute inflammation is responsible for most alterations in intestinal functions that often persist following the resolution of the acute inflammation process and are frequently observed during remission in patients with IBD. Currently, there are no effective treatments for UC that prevent the inflammation-related acute and long-term intestinal dysfunctions, but only combined pharmacological and nutritional therapies, which encompass assessment of daily caloric intake and periodic measurement of functional capacities. 5-aminosalicylic acid is the first therapeutic approach with corticosteroids and thiopurines. However, the high incidence of side effects, including hypersensitivity reactions, liver toxicity, immunosuppression, and pancreatitis [5] associated with the increasing number of corticosteroids/thiopurines-resistant UC led to the development of alternative therapeutical strategies that include anti-TNF-α, anti-IL-12/23 p40, and anti-integrin or JAK inhibitors [6]. Biological therapies, such as infliximab and adalimumab, also have limitations mainly related to the reduced compliance for parenteral administration, immunogenicity, and high costs that reduce their clinical application. N-palmitoyl-D-glucosamine (PGA) is a natural amide of palmitic acid and glucosamine that shares the anti-inflammatory properties with its endogenous analog palmitoylethanolamide (PEA) and those of glucosamine [7]. PGA belongs to the ALIAmide family (ALIA, autacoid local injury antagonism), a class of both synthetic and endogenous fatty acid amides that display a wide range of homeostatic effects in response to increased oxidative stress and cell damage, including anti-inflammatory and pain-relieving effects [8]. In particular, PEA has repeatedly been shown to improve clinical and histological signs of colitis in different murine models [9,10]. In line, micronized PGA resulted in being safe (LD50 ≥ 2000 mg/kg) and effective in preclinical models of inflammation and osteoarthritis (OA) pain, taking advantage of particle size reduction that enhances its anti-inflammatory activity [7]. To note, the anti-inflammatory action of PGA might derive from both the amide and monosaccharide portions [11]. In fact, glucosamine improved colitis symptoms in DSS-treated mice by preventing intestinal epithelial cell activation and tight junction proteins expression decrease, with a parallel decrease in the nuclear factor-kappa B (NF-kB) activity and reduced TNF-α and IL-1β release [11,12]. Moreover, oral glucosamine and chondroitin compositions positively impact microbiota composition in the intestine of healthy adults, supporting PGA similar acitivity [13]. Iannotta et al. also reported that micronized PGA acts as a toll-like receptor (TLR)-4 antagonist, thanks to its structural similarity to the lipid A component of lipopolysaccharide (LPS). TLR-4 is involved in activating the nucleotide-binding oligomerization domain leucine-rich repeat and pyrin domain-containing protein 3 (NLRP3) inflammasome complex, which is overactivated in several inflammatory syndromes, including intestinal inflammation, neurodegenerative, and metabolic diseases [14]. NLRP3 inflammasome plays a critical role in inflammatory response as a major component of innate immunity and is involved in exacerbating the mucosal immune response and intestinal epithelial barrier damage during colitis [15]. ALIAmide-mediated activation of the peroxisome proliferator-activated receptors-α (PPAR-α) counters the NLRP3 activity [16], although whether PGA exerts its anti-inflammatory effects via PPAR-α receptors is still unknown.

In this study, we investigated the effectiveness and the mechanisms of action of micronized PGA in a murine model of colitis induced by the 2,4-dinitrobenzene sulfonic acid (DNBS) and assessed the in vivo effects of orally administered micronized PGA on (i) colitis severity, (ii) mucosal inflammation and immune cells infiltration, (iii) release of pro-inflammatory cytokines, and (iv) NLRP3 and TLR-4 intestinal activation in a resolution phase of intestinal inflammation.

## 2. Materials and Methods

### 2.1. Animals and Experimental Design

Eight-week-old male C57BL/6J mice (Charles River, Lecco, Italy) (*n* = 50) have been used for the experiments. The procedures included in the experimental plan have been approved by Sapienza University’s Ethics Committee. Animal care followed the (International Association for the Study of Pain) IASP and European Community (EC L358/1 18/12/86) guidelines on the use and protection of animals in experimental research. Colitis groups received a single intracolonic administration of 4 mg DNBS (Sigma Aldrich, St. Louis, MO, USA) in 100 μL of 50% ethanol (Sigma Aldrich, St. Louis, MO, USA) and saline (Thermo Fisher Scientific, Waltham, MA, USA), whereas the vehicle group received a single intracolonic administration of saline and ethanol as described below. Overnight-fasted mice were treated with DNBS on day 0 through a soft cannula (Hugo-Sachs Elektronik, March, Germany) quickly inserted around 3cm away from the anus without anesthetization. DNBS solution was slowly administrated into the colon-rectal tract, and animals were maintained slightly sloped for the entire procedure. Thus, mice were placed back into their cages and kept overnight on a heating pad to aid recovery. Disease activity index (DAI) parameters were daily recorded from day 0 to day 7 to assess colitis severity. Micronized PGA was suspended in carboxymethylcellulose (CMC) (Thermo Fisher Scientific, Waltham, MA, USA) and 1X PBS (Sigma Aldrich, St. Louis, MO, USA), and 200 µL of 30 mg/kg and 100 mg/kg PGA suspension were daily given by a single gavage from days 1 to 6 based on the experimental design [7]. PGA was used in the micronized formulation kindly provided by Epitech Group S.p.A (Saccolongo, Italy). Micronized formulation of PGA with a smaller particle size overcomes the low water-solubility issue of this compound, increasing its oral absorption and bioavailability. In fact, drug solubility is strongly related to particle size and its reduction leads to an increase in the specific surface area with enhanced solubility and potentially higher bioavailability. PPAR-α antagonist MK886 (Selleck Chemicals, Houston, TX) was dissolved in 1X PBS and given daily at the dose of 10 mg/kg by a single intraperitoneal (IP) administration (150 µL) from days 1 to Mice were randomly divided into the following groups (*n* = 10 each): (1) vehicle group receiving single intracolonic administration of saline; (2) colitis group; (3) colitis group receiving a daily gavage with 30 mg/kg micronized PGA; (4) colitis group receiving a daily gavage with 100 mg/kg micronized PGA; (5) colitis group receiving a daily gavage with 100 mg/kg micronized PGA associated with daily intraperitoneal administration of 10 mg/kg PPAR-α antagonist MK886. Animals in the vehicle and colitis groups also received a daily gavage of CMC solution in 1X PBS (200 µL) and a daily IP with 1X PBS (150 µL) from days 1 to day 6. Mice were euthanized on day 7 by cervical dislocation; thus, spleen weight and colon length were measured, and blood samples, as well as colon tissues, were collected to conduct histochemical and biochemical analyses as described below. Each experimental group included *n* = 10 mice. All the experiments were performed in triplicate on the distal colon by randomly using N = 5 colon for histological staining and N = 5 colon for immunofluorescence analysis.

### 2.2. Disease Activity Index (DAI)

The DAI score was used to evaluate the colitis severity and progression during the 7 days of the experimental protocol, according to the criteria developed by Cooper et al. [17]. The scored parameters were: (i) changes in body weight; (ii) stool consistency; (iii) rectal bleeding. DAI score was recorded daily (from day 0 to 7), scores were given depending on the severity of the symptoms, and the results were expressed as cumulative average scores in each experimental group.

### 2.3. Histopathological Analyses

Colonic tissues were fixed in 4% paraformaldehyde (PFA) (Thermo Fisher Scientific, Waltham, MA, USA) and cryo-sectioned in 15 μm slices. Slices were stained with hematoxylin and eosin (H&E) (Sigma Aldrich, St. Louis, MO, USA) to evaluate the histopathological damage score according to Li et al. [18]. Criteria included: (i) distortion and loss of crypt architecture; (ii) inflammatory cells infiltration; (iii) muscle thickening; (iv) goblet cells depletion; (v) crypts absence.Slices were analyzed with a microscope Nikon Eclipse 80i by Nikon Instruments Europe (Nikon Corporation, Tokyo, Japan), and images were captured at 10× magnification by a high-resolution digital camera (Nikon Digital Sight DS-U1). Cumulative damage scores obtained from each experimental group were expressed as average scores.

### 2.4. Immunofluorescence Analysis on Colonic Sections

Immunofluorescence analyses were performed on 15 µm colonic slices fixed in ice-cold 4% PFA. Sections were blocked with a solution composed of 1X PBS, 4% Normal Donkey Serum, 0.4% (Merk Millipore, St. Louis, MO, USA) TRITON-100 (Sigma Aldrich, St. Louis, MO, USA), and 1% Bovine Serum Albumin (BSA) (Sigma Aldrich, St. Louis, MO, USA) for 45 min and subsequently incubated at +4 °C overnight with primary antibody (Table 1). Slices were then washed with 1X PBS and incubated in the dark at +4 °C with fluorescein isothiocyanate-conjugated anti-rabbit (1:1000 dilution *v*/*v*; Abcam, Cambridge, UK) or Texas Red-conjugated anti-mouse (1:500 dilution *v*/*v*, mouse; Abcam, Cambridge, USA). Sections were analyzed with a microscope Nikon Eclipse 80i, and images were captured at 20× and 40× magnification by a high-resolution digital camera (Nikon Digital Sight DS-U1). Results were expressed as relative fluorescence units (RFU) and fluorescence intensity percentage (FI%).

### 2.5. Enzyme-Linked Immunosorbent Assay (ELISA) for IL-1β and PGE_2_

Enzyme-linked immunosorbent assay (ELISA) for PGE_2_ and IL-1β (Thermo Fisher Scientific, Waltham, MA, USA) was carried out on mouse plasma isolated from blood samples according to the manufacturer’s protocol. Absorbance was measured on a microtiter plate reader. PGE_2_ and IL-1β levels were determined using standard curve methods.

### 2.6. Statistical Analyses

Results were expressed as the mean ± SD. Statistical analysis was performed using parametric one-way analysis of variance (ANOVA) and multiple comparisons were performed by Bonferroni’s post hoc test. *p*-values < 0.05 were considered statistically significant. Data were analyzed by using Graphpad Prism and ImageJ software.

## 3. Results

### 3.1. Micronized PGA Improves the Disease Spectrum and Macroscopic Signs of Colitis in a Dose-Dependent Manner through PPAR-α Involvement

The DAI score was significantly increased in the colitis group during the 7 days that followed DNBS administration (7.9 ± 0.141, *p* < 0.0001; Figure 1A), with a parallel colonic shortening (3.83 ± 0.543 cm, *p* < 0.0001; Figure 1B,D) and spleen weight increase (0.153 ± 0.0269 g, *p* < 0.05; Figure 1C) in comparison to the vehicle group. PGA resulted in a dose-dependent improvement of overall colitis hallmarks, leading to a significant decrease in DAI score (4.65 ± 0.354, *p* < 0.001 and 3.07 ± 0.305, *p* < 0.0001 for the 30 mg/kg and 100 mg/kg dose, respectively; Figure 1A), increase in colon length (5.82 ± 0.676 cm and 7.55 ± 0.572 cm, *p* < 0.0001 for 30 mg/kg and 100 mg/kg dose, respectively; Figure 1B), and reduction of spleen weight (0.0897 ± 0.011 g, *p* < 0.05 for 100 mg/kg m-PGA, Figure 1C) as compared to DNBS-treated mice. DAI score, colon length, and spleen weight were comparable to those in DNBS mice treated with 100 mg/Kg um-PGA and PPAR-α antagonist MK886 (10 mg/kg), suggesting that PGA exerts its beneficial effects on colitis through the selective involvement of PPAR-α receptors.

### 3.2. Micronized PGA Ameliorates Mucosal Integrity and Prevents Colonic Histological Damage in DNBS-Treated Mice

DNBS-treated mice maintained a significant impairment of colonic mucosal integrity on day 7 following colitis induction, as demonstrated by the decreased expression of the two tight junction proteins, zonula occludens-1 (ZO-1) and occludin, compared to the vehicle group (25.08 ± 3.196 FI%, *p* < 0.0001 for ZO-1 and 18.23 ± 1.322 FI%, *p* < 0.0001 for occludin; Figure 2A–C). The loss of ZO-1 and occludin was partially prevented by PGA at 30 mg/kg dose (37.4 ± 2.507 FI%, *p* <0.0001 for ZO-1 and 33.97 ± 2.217 FI%, *p* < 0.0001 for occludin; Figure 2A–C), whereas a marked restoration was observed in colitis mice treated with 100 mg/kg PGA in comparison to the DNBS group (87.8 ± 2.579 FI% and 94.75 ± 3.988 FI%, *p* < 0.0001 for ZO-1 and occludin, respectively; Figure 2A–C). The effects of PGA (100 mg/kg) were reverted by MK886 (10 mg/kg) in colitis mice, further demonstrating the involvement of PPAR-α receptors. Histopathological scores revealed extensive damage in the colonic mucosal barrier of DNBS-treated mice with marked neutrophil infiltration and damaged mucosal integrity compared to vehicle (8.2 ± 0.9189, *p* < 0.0001; Figure 2D,E). Micronized PGA (30 mg/kg) preserved mucosal integrity and counteracted the neutrophil infiltration within the mucosa, although the number of crypts was lower in comparison to the DNBS group (5.5 ± 0.9718, *p* < 0.0001; Figure 2D,E). At the higher dose (100 mg/kg), PGA widely restored mucosal integrity, significantly reducing the neutrophil infiltration, and preserving the architecture and number of the crypts in comparison to the DNBS group (2.5 ± 0.8498, *p* < 0.0001; Figure 2D,E). Co-administration of PGA and MK886 resulted in colonic histological damage comparable to DNBS-group, further supporting that PGA preserves the intestinal mucosal integrity by PPAR-α engagement.

### 3.3. Micronized PGA Downregulates the TLR-4/NLRP3/iNOS Expression and Decreases the Release of Inflammatory Mediators in DNBS-Treated Mice via PPAR-α Receptors

Our results show that TLR-4, NLRP3, and inducible nitric oxide synthase (iNOS) expression was markedly increased within the mucosa following DNBS-induced colitis compared to the vehicle (31.38 ± 1.169 RFU, 14.54 ± 1.282 RFU, and 33.93 ± 1.665 RFU for TLR-4, NLRP3 and iNOS, respectively; *p* < 0.0001 for all comparisons; Figure 3A–D). In DNBS mice receiving 30 mg/kg PGA, the expression of TLR-4, NLRP3, and iNOS was significantly reduced (21.08 ± 1.904 RFU, 10.40 ± 1.235 RFU, and 18.26 ± 1.668 RFU for TLR-4, NLRP3 and iNOS, respectively; *p* < 0.0001 for all comparisons; Figure 3A–D) and completely downregulated by PGA 100 mg/kg (14.98 ± 0.712 RFU, 7.54 ± 0.537 RFU, and 12.56 ± 1.105 RFU for TLR-4, NLRP3, and iNOS, respectively; *p* < 0.0001 for all comparisons; Figure 3A–D). According to previous results, 100 mg/kg PGA did not show any effect in DNBS mice co-administered with PPAR-α antagonist MK866 (10 mg/kg), supporting that PGA prevents the activation of the TLR4/NLRP3/iNOS pathway through the PPAR-α-mediated mechanism. Further, the increased plasma levels of IL-1β and PGE_2_ detected in DNBS mice at day 7 after colitis induction (113.1 ± 32.71 pg/mL and 348.5 ± 45.22 pg/mL for IL-1β and PGE_2_, respectively; *p* < 0.0001 vs. vehicle for both comparisons; Figure 3E,F) were significantly and dose-dependently decreased by PGA (62.1 ± 15.09 pg/mL and 36.7 ± 13.12 pg/mL for IL-1β in the lower and higher dose group, respectively; 217.5 ± 61.07 pg/mL and 169.7 ± 37.12 pg/mL for PGE_2_ in the lower and higher dose group, respectively; *p* < 0.0001 vs. DNBS for all comparisons; Figure 3E,F). The anti-inflammatory effect of PGA was significantly inhibited in the presence of PPAR-α antagonist MK886, displaying comparable plasma cytokines levels with the DNBS group.

## 4. Discussion

IBD showed an increasing trend of incidence worldwide, and current therapies mainly consist of the chronic administration of immunosuppressive drugs [19]. However, these drugs display a short-term efficacy, and they are not suitable as a maintenance therapy due to related systemic adverse reactions [20]. The possibility of exploiting new compounds to target different stages and pathways of the inflammatory response represents an outstanding challenge for IBD therapy. In the present study, we provided evidence that micronized PGA significantly and dose-dependently counteracts the occasional inflammatory-mediated intestinal functions alteration during the colitis resolution phase that characterizes the remission state of the disease. 

Oral administration of micronized PGA improved the DAI score and resulted in macroscopic amelioration of intestinal inflammation, as shown by the increased expression of tight junction proteins (ZO-1 and occludin) in the colonic mucosa and lower histological damage score. Similar to the homologous ALIAmide PEA, which was successful in several colitis models treatment in mice and humans [21,22,23,24,25], PGA revealed an anti-inflammatory effect through a PPAR-α-dependent mechanism since in the presence of PPAR-α antagonist MK886, PGA-mediated effects were abolished. We demonstrated that PGA caused a significant decrease of pro-inflammatory mediators, such as iNOS and NLRP3 protein expression in colonic tissues as well as PGE_2_ and IL-1β cytokines release in mice plasma. This anti-inflammatory effect of PGA might be related to its monosaccharide portion, glucosamine. In support, glucosamine was able to reduce colitis-associated symptoms in DSS-treated mice by reducing TNF-α, IL-1β, and NF-kB expression in the colonic mucosa, and increasing the expression of ZO-1 and occludin tight junction proteins [12]. Patients recovering from an exacerbation of ulcerative/membranous colitis display increased levels of glucosamine synthetase. This enzyme synthesizes N-acetyl-glucosamine (NAG), which was recently implicated in the tissue regeneration phase in pediatric IBD [26,27,28]. In addition, glucosamine and/or chondroitin are used as dietary supplements to reduce colonrectal cancer risk and systemic inflammation [29,30,31,32]. This evidence supports the hypothesis that glucosamine might be responsible, at least in part, for PGA anti-inflammatory activity once hydrolyzed by the degradative enzymes (FAAH, NAAA), although further confirmation is required.

In agreement with previous studies demonstrating that PGA negatively regulates TLR-4 signaling enhanced by intestinal inflammation and neuropathic pain [8,14], our results show that micronized PGA dose-dependently reduced TLR-4 expression in the colonic mucosa of colitis mice, suggesting that this may contribute to its potent anti-inflammatory activity. The sequel of events triggered by TLR-4 activation, including NF-kB and NLRP3 activation, and the release of IL-1β, IL-6, and TNF-α, are considered the most involved in persistent intestinal inflammation triggering and maintenance during colitis [8,14], and HIV-1 Tat-induced diarrhea [33]. In this context, PPAR-α agonists, such as PEA, may mediate TLR-4 down-regulation and efficiently suppress the inflammatory process similar to the evidence observed in vitro [15], clinical studies [34], endotoxin induced-uveitis rat model, and DSS-induced colitis mice model [21,35]. 

In addition to PPAR-α agonism, PGA displays structural similarity with the lipopolysaccharide (LPS) component Lipid A, which results in a direct antagonism on TLR-4 [14]. Thus, the anti-inflammatory properties of PGA might be related to two distinct mechanisms, although this hypothesis requires further investigation.

Previous studies have shown that PPAR-α activation inhibits immune cell infiltration in colonic mucosa and decreases the expression and release of pro-inflammatory markers in mice and humans [36,37]. In line, we have shown that micronized PGA reduces immune cell infiltration within the colonic mucosa and downregulates NLRP3 inflammasome expression via PPAR-α activation. This anti-inflammatory activity probably allows PGA to preserve the intestinal epithelial barrier integrity. Actually, NLRP3 plays a key role in exacerbating the mucosal immune response and intestinal epithelial barrier damage [38]. The activation of PPAR-α receptors prevents NLRP3 hyper-activation, which is linked to several inflammatory syndromes, including intestinal inflammation [36]. Here, we demonstrated the efficient and dose-dependent counteraction of NLRP3 overexpression by oral administration of micronized PGA. NLRP3 inhibition was associated with reduced colitis severity and histological damage of colonic mucosa, supporting that PGA exerts an anti-inflammatory effect by suppressing NLRP3 activity through PPAR-α receptor involvement.

To gain more mechanistic insights, we evaluated the effect of micronized PGA on TLR-4 expression, which plays an essential role in innate immunity activation by recognizing microbial antigens, such as LPS [39]. The increased translocation of luminal microbe-derived products that follows the epithelial barrier breakdown during intestinal inflammation leads to higher TLR-4 activation. The downstream sequel events triggered by the TLR-4 activation, including NLRP3 overexpression, iNOS upregulation, and increased release of pro-inflammatory cytokines such as IL-1β and PGE_2_ [40,41], are strictly related to mucosa damage expansion [42], and visceral hypersensitivity associated with the inflammatory process [43,44]. Oral administration of micronized PGA resulted in decreased expression of TLR-4 and related downstream pro-inflammatory pathways in DNBS-treated mice, as demonstrated by the parallel downregulation of iNOS and NLRP3 expression, as well as by the reduced release of IL-1β and PGE_2._ This suggests that PGA might act as a safe “multitarget” modulator of intestinal inflammation and prevents the long-term intestinal dysfunction that generally follows the acute phase of colitis.

To note, TLR-4 activation is also associated with visceral hypersensitivity and related behavioral disorders, resulting in episodes of anxiety and recurrent flares of abdominal pain [45,46,47]. This latter was also associated with long-term changes in the intestinal microbiota composition, pointing out that compounds able to restore and maintain microbiota homeostasis might provide benefit in IBD patients with recurrent visceral allodynia and hyperalgesia [48]. Interestingly, a bacteria strain derived from legumes is able to produce PGA (*rhizobium leguminosarum*) [49], and successfully colonize the intestinal microenvironment by acting as a xenobiotic metabolizer [47]. Taking the evidence that oral glucosamine and chondroitin compositions modulate positively the intestinal microbiota in healthy adults [13], micronized PGA might have the capability to target distinct pathological aspects of intestinal inflammation by downregulating pro-inflammatory mediators, decreasing mucosal damage and visceral hypersensitivity, and even restoring microbiota homeostasis.

In addition, antagonism on TRL-4 could potentially determine the regulation of mucosal blood flow since a reduced expression of TRL-4 and consequent down-regulation of NF-kB leads to inactivity of iNOS and reduced release of nitric oxide (NO). Adequate organ blood flow also demonstrates a protective and healing-promoting role in the gastrointestinal tract [50,51,52]. Despite further studies being needed to confirm this hypothesis, our results provide the first evidence on the ability of micronized PGA to target colitis through a double mechanism of action: PPAR-α agonism and TLR-4 antagonism [14]. In consideration of the obtained results and the safety profile of micronized PGA, more studies are advised to explore the protective effects of micronized PGA in IBD management.

## Figures and Tables

**Figure 1 biomolecules-12-01163-f001:**
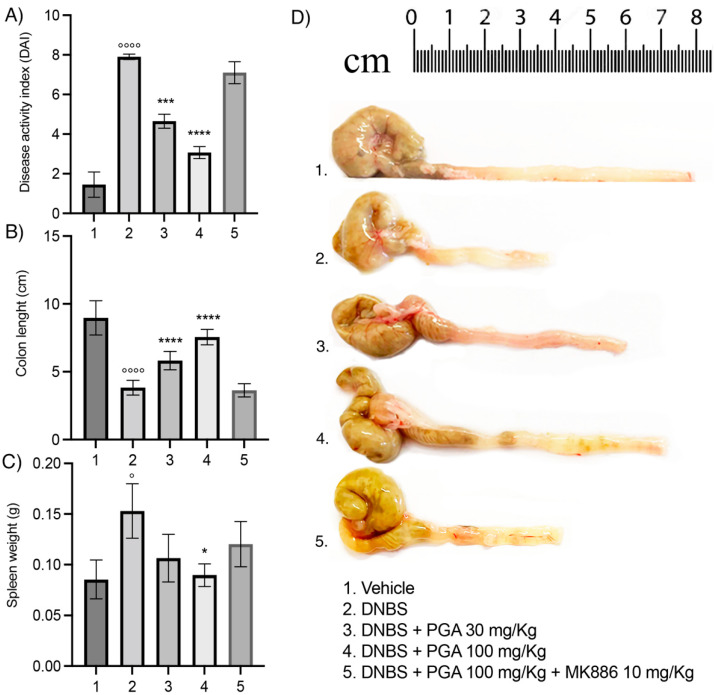
Micronized PGA significantly improves colitis hallmarks in PPAR-α-dependent manner. The effects of oral administration of micronized PGA on (**A**) DAI score, (**B**,**D**) colonic length, and (**C**) spleen weight in DNBS-treated mice. Results are expressed as the mean ± SD of n = 5 experiments ° *p* < 0.05 vs. vehicle; °°°° *p* < 0.0001 vs. vehicle; * *p* < 0.05 vs. DNBS; *** *p* < 0.001 vs. DNBS; **** *p* < 0.0001 vs. DNBS. Peroxisome proliferator-activated receptors-α (PPAR-α); Disease Activity Index (DAI); 2,4-dinitrobenzene sulfonic acid (DNBS).

**Figure 2 biomolecules-12-01163-f002:**
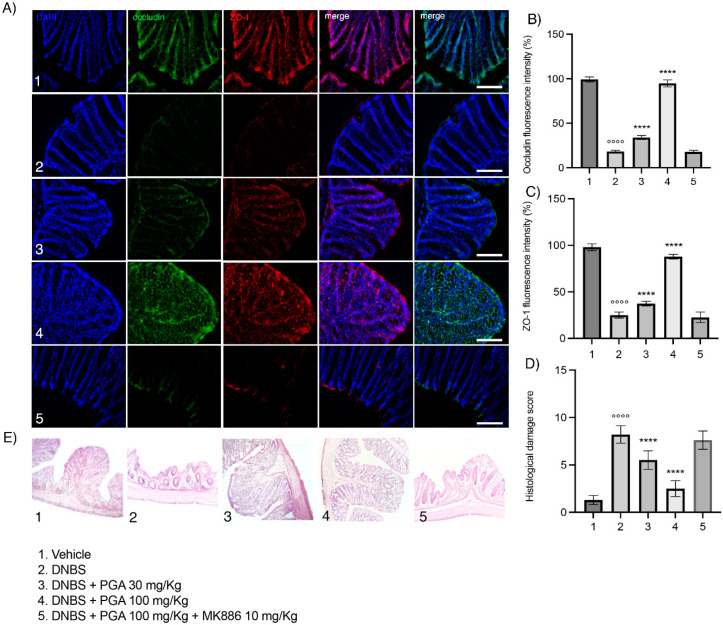
Micronized PGA prevents the loss of tight junction proteins ZO-1 and occludin, and colonic barrier disruption through PPAR-α involvement. (**A**) Representative images show double-label immunohistochemistry for occludin (green) and ZO-1 (red) in the colon with (**B**,**C**) relative quantification. Nuclei were also labeled by DAPI. (**E**) Representative images of H&E-stained on distal colon sections, and (**D**) relative histological damage score. Results are expressed as the mean ± SD of n = 5 experiments (25 slices for each animal). °°°° *p* < 0.0001 vs. vehicle; **** *p* < 0.0001 vs. DNBS. Scale bar = 100 μm; magnification 20×. Zonula occludens (ZO-1); peroxisome proliferator-activated receptors-α (PPAR-α); hematoxylin and eosin (H&E), 2,4-dinitrobenzene sulfonic acid (DNBS).

**Figure 3 biomolecules-12-01163-f003:**
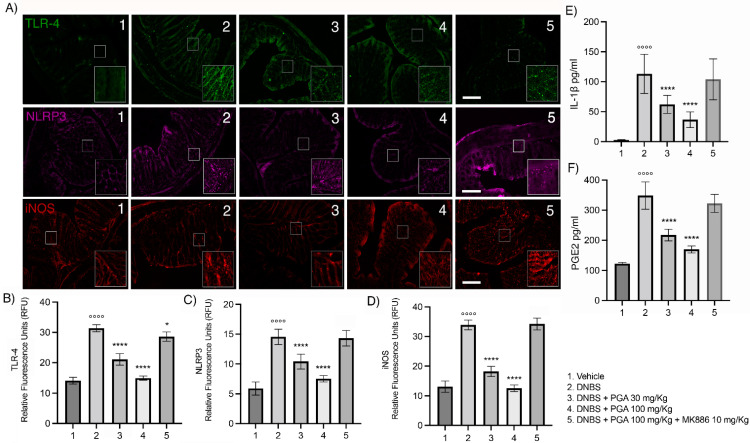
Micronized PGA decreases TLR-4/NLRP3/iNOS expression in mice colon and pro-inflammatory cytokines release in plasma samples throughout selective PPAR-α involvement in DNBS-treated mice. (**A**) Immunofluorescence images display TLR-4 (green), NLRP3 (magenta), and iNOS (red) staining, and the relative quantification for (**B**) TLR-4, (**C**) NLRP3, and (**D**) iNOS. (**E**) IL-1β and (**F**) PGE_2_ release in mice plasma. Results are expressed as the mean ± SD of n = 5 experiments (25 slices for each animal). °°°° *p* < 0.0001 vs. vehicle; * *p* < 0.05 vs. DNBS; **** *p* < 0.0001 vs. DNBS. Scale bar = 100 μm; magnification 20× 40×. Toll-like receptor (TLR)-4; nucleotide-binding oligomerization domain leucine-rich repeat and pyrine domain-containing protein 3 (NLRP3); inducible nitric oxide synthase (iNOS); peroxisome proliferator-activated receptors-α (PPAR-α); interleukin-1β (IL-1β); prostaglandin E2 (PGE_2._), 2,4-dinitrobenzene sulfonic acid (DNBS).

**Table 1 biomolecules-12-01163-t001:** Primary antibodies used in immunofluorescence analyses on cryo-sectioned colon slides.

Antibody	Host	Clonality	Dilution	Brand
ZO-1	Mouse	Monoclonal	6 microgram *w*/*v*	Invitrogen, Thermo Fisher, Waltham, MA, USA
Occludin	Rabbit	Polyclonal	1:100 *v*/*v*	Bioss Antibodies, Boston, MA, USA
NLRP3	Rabbit	Polyclonal	1:1000 *v*/*v*	Invitrogen, Thermo Fisher, Waltham, MA, USA
TLR-4	Rabbit	Polyclonal	1:150 *v*/*v*	Bioss Antibodies, Boston, MA, USA
iNOS	Mouse	Monoclonal	1:1000 *v*/*v*	Novusbio, Centennial, CO, USA

Zonula occludens (ZO-1); nucleotide-binding oligomerization domain leucine-rich repeat and pyrine domain-containing protein 3 (NLRP3); toll-like receptor (TLR)-4; inducible nitric oxide synthase (iNOS).

## Data Availability

The data presented in this study are available on request from the corresponding author. The data are not publicly available due to the policy of our research group, but we will share data unreservedly on online data sharing platforms upon request.

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
