# Peer review of "N-Palmitoyl-D-Glucosamine Inhibits TLR-4/NLRP3 and Improves DNBS-Induced Colon Inflammation through a PPAR-α-Dependent Mechanism"

_biomolecules, 2022, doi:10.3390/biom12081163_

Round 1

Reviewer 1 Report

Even with vast improvements in IBD therapeutics in the past few decades, there still remains an unmet clinical need to provide therapies with improved efficacy. The biggest problem at hand remains the maintenance of prolonged durable remission and loss of response in patients. This paper presents an exciting prospect to utilize micronized PGA as a means to ameliorate immuno-inflammatory responses to prevent long-lasting effects in the intestinal epithelium.  I do have a few queries and suggestions as below:

1. Please describe some details for readers as to why micronized PGA is considered best considering its particle size reduction and anti-inflammatory effects.

2. Please mention the Disease activity index and histopathological scores in a more organized manner. The current presentation makes it cluttered and confusing to maneuver around.

3. Can you please describe the rationale behind 30mg/kg and 100 mg/kg of PGA? Why was an intermediate dose not kept as a group in the study?

4. For the assessment of colitis in mice in this study, would the assessment of peritoneal fat index have further added to your results? 

5. The number of crypts was lower in the 30mg/kg PGA group, can you please describe how that maybe be significant?

Author Response

# REVIEWER 1:

Even with vast improvements in IBD therapeutics in the past few decades, there still remains an unmet clinical need to provide therapies with improved efficacy. The biggest problem at hand remains the maintenance of prolonged durable remission and loss of response in patients. This paper presents an exciting prospect to utilize micronized PGA as a means to ameliorate immuno-inflammatory responses to prevent long-lasting effects in the intestinal epithelium.  I do have a few queries and suggestions as below:

Thank you for your very kind and constructive feedback. We are pleased that the reviewer felt that the initial draft contained important results and we appreciate the reviewer’s comments on how to improve the presentation and content. We have made all suggested revisions and addressed each specific point raised by the reviewer as indicated below.

1. Please describe some details for readers as to why micronized PGA is considered best considering its particle size reduction and anti-inflammatory effects

This is a good point and we agree that mentioning the impact of micronization on PGA bioavailability is important. We added text for this area to the “Animal and experimental design” section (P4 line 151).

2. Please mention the Disease activity index and histopathological scores in a more organized manner. The current presentation makes it cluttered and confusing to maneuver around.

We agree with the reviewer and we modified the sentences as suggested (P4 line 172, P5 line 182).

3. Can you please describe the rationale behind 30mg/kg and 100 mg/kg of PGA? Why was an intermediate dose not kept as a group in the study?

Yet little is known regarding the therapeutic window of micronized PGA in the treatment of intestinal inflammation. Thus, we considered the same therapeutic range used by Cordaro et al. in preclinical models of inflammation and osteoarthritis pain [PMID: 31779692]. We cited the reference above to reflect the content of our choice (line 145, ref. 7).

4. For the assessment of colitis in mice in this study, would the assessment of peritoneal fat index have further added to your results? 

This is a good point and we thank the reviewer for their suggestion. Abdominal fat type and distribution have been associated with complicated IBD and adverse postoperative outcomes. A higher degree of abdominal adiposity and a lower skeletal mass can be expected in IBD patients, although DNBS-treated mice do not display significant alteration in their abdominal fat degree after 7 days. For that reason, we considered parameters, such as colon length and spleen weight, which are usually altered in the DNBS-induced colitis model in mice.

5. The number of crypts was lower in the 30mg/kg PGA group, can you please describe how that maybe be significant?

A decrease in crypts' number occurs during inflammation because of alteration of their homeostasis mediated by immune cells that penetrate within the intestinal mucosa. Colitis mice treated with 30 mg/kg PGA displayed a mild restoration of mucosal crypt number in comparison with those treated with 100 mg/kg PGA likely related to the concentration-dependent effect of the PGA.

Reviewer 2 Report

This paper examines the effect of orally administered PGA on DNBS induced colon injury for gross protection and some detail on molecular levels effects and involvement of PPAR-alpha and TLR-4/NLRP3/iNOS, the later by mRNA levels.  Plasma IL-1 and PGE2 beta were measured, too. 

The results, by themselves, are interesting and show a real impact of micronized PGA on relieving the damaging effect of DNBS.  The problem is one of perspective.  Is the effect of PGA any better than that of PEA or is the use of PGA based on greater ease and/or efficiency of delivery or bioavailability  than PEA?  Glucosamine is indicated to add to the effect of the compound in a gross manner very close the effect of PGA based on reference 11, which makes it harder to know whether PGA is really superior to either PEA or glucosamine.  It would add much value to the paper, if the study was expanded to include PEA and glucosamine separately.

Minor- add a couple of sentence on how micronization adds to the effect of PGA.

Author Response

# REVIEWER 2:

This paper examines the effect of orally administered PGA on DNBS induced colon injury for gross protection and some detail on molecular levels effects and involvement of PPAR-alpha and TLR-4/NLRP3/iNOS, the later by mRNA levels.  Plasma IL-1 and PGE2 beta were measured, too.

The results, by themselves, are interesting and show a real impact of micronized PGA on relieving the damaging effect of DNBS.

We are pleased that the reviewer felt that the initial draft contained important information and we appreciate the reviewer’s comments on how to improve the presentation and content. We have made all suggested revisions and addressed each specific point raised by the reviewer as indicated below.

1. Is the effect of PGA any better than that of PEA or is the use of PGA based on greater ease and/or efficiency of delivery or bioavailabilitythan PEA?

This is a very important point, although little is known regarding the different efficiency between PEA and PGA in the context of intestinal inflammation. To date, we know that PGA exhibits comparable pharmacokinetic to PEA, while only Peritore et al. has displayed that orally administrated um-PEA was effective in counteracting DNBS-induced colitis in mice (PMID: 33809584). However, a consistent comparison between PEA and PGA regarding their positive effects during colitis is not possible because of differences in dosages, level of micronization, and diverse inflammatory endpoints studied.

2. Glucosamine is indicated to add to the effect of the compound in a gross manner very close the effect of PGA based on reference 11, which makes it harder to know whether PGA is really superior to either PEA or glucosamine.It would add much value to the paper, if the study was expanded to include PEA and glucosamine separately.

This is a very important point, and we agree that investigating which portions of the PGA molecule are responsible for its anti-inflammatory properties might provide essential information and drive its more rational use during inflammatory conditions in the intestine. However, this expands the scope of our initial investigation which aimed to test whether micronized PGA was efficient in reducing the colitis severity and preventing the inflammatory-related changes in the intestinal function that persist in the resolution phase of colitis. The encouraging results that we obtained in this study will drive our further investigations of PGA to deepen its mechanism of action in comparison with PEA and glucosamine in the same experimental conditions.

3. add a couple of sentence on how micronization adds to the effect of PGA.

Thank you for this suggestion. We have added this information as suggested (P4 line 151).

Reviewer 3 Report

Manuscript # biomolecules-1853993

Title: N-palmitoyl-D-glucosamine inhibits TLR-4/NLRP3 and improves DNBS-induced colon inflammation through a PPAR-alpha-dependent mechanism

Authors: Irene Palenca et al.

The above manuscript is interesting; the authors tested the influence of N-palmitoyl-D-glucosamine, a natural amide of palmitic acid and glucosamine, given after rectal application of 2,4-dinitrobenzene sulfonic acid (DNBS) on the development of DNBS-induced colitis. However, it should be noted that the manuscript contains some deficiencies and errors that must be removed before accepting the manuscript for publication.

List of the errors and deficiencies:

  1. Abstract, line 30. What does “improves colitis severity” mean? Replace “improves” with “reduces”.
  2. Abstract, line 30. The authors should clearly state what kind of “alterations of intestinal barrier function” was prevented. This part of the sentence is unclear, the more so as the next sentence addresses the problem more precisely. Therefore, it would be advisable to omit this part of the sentence and also modify the beginning of the next sentence.
  3. Section “2.1. Animals and experimental design”, line 108. The authors should present the total number of animals used in the study.
  4. Line 110 and next lines. All abbreviations should be presented in their full name at the point where they appear for the first time. Full names of abbreviation should be repeated in the body of the manuscript at the place of the first use, as well as in tables and Figure legends. Tables and Figures should be understandable without carefully reading the text of the manuscript. On the other hand, the number of abbreviations should be reduced. The excess of abbreviations makes the manuscript difficult to understand.
  5. Line 113. What does “saline water 0.9%” mean? It should be replaced with 0.9% NaCl solution or simply saline.
  6. Section 2.1. Animals and experimental design, lines 113-118. The text is messy. Each experimental group should be described separately. The authors should write that colitis was induced by intracolonic administration of DNBS. The method should be described in detail. Has colitis induction been performed in conscious or anesthetized animals? If animals were anesthetized, how was this done? How deep was the colon catheter inserted? Were animals fasted before? What was the reason for choosing the GA and MK886 doses used? Were the daily doses administered in one single dose or divided doses? In what form were PGA and MK886 served? What was obtained, solution or suspension? What was the vehicle for compounds used? What was the volume of doses administered? Were animals in control group and colitis group without administration of test agents receiving placebo (vehicle) by gavage and intraperitoneally in the volume and at the same time as the animals receiving PGA and MK886? Were the animals receiving PGA without MK886 administered anything intraperitoneally? All this information should be included in the manuscript.
  7. Section 2.1. Animals and experimental design, lines 120-121. The authors stated that “Mice were allowed to recover for 7 days following DNBS administration”. The animals were killed on the 7th day and the authors did not check histological and biochemical parameters of the severity of colitis. For this reason, the authors had no information that there was any recovery between the firs and the seventh day of the study.
  8. Section 2.1. Animals and experimental design, line 123. How were animals euthanized? The authors should present the method of euthanasia in detail.
  9. Lines 61-62. In the case of all drugs, regents, assays and equipment, the authors should provide general and trade name of the reagent or equipment, manufacturer’s name, city, and country.
  10. In the section 2.1. Animals and experimental design. The authors stated that they used 10 animals in each experimental group. In contrast, In Figure 1 and 3, the authors wrote that n was 5. What was the reason of these discrepancies? Was any premature animal mortality or selection of results prior to statistical analysis? This should be clearly stated in the manuscript. In the case of Figure 2 B C and D, the number of animals/observations is not known.
  11. The authors stated in the manuscript that “Yomogida et al. demonstrated the in-vivo suppressive effect of glucosamine on intestinal epithelial cell activation in a model of DSS-induced colitis, suggesting that the anti-inflammatory action of PGA might derive from both the amide and monosaccharide portions [11]”. The relationship between glucosamine, experimental colitis, and IBD has been studied much more widely. The observations of Yomogida et al. have been confirmed by Bak et al. (PMID: 24325781). They have found that glucosamine reduces colitis-associated symptoms in DSS-induced colitis. This effect was associated with reduction in Tumor necrosis factor-α (TNF-α), interleukin-1β, and nuclear factor-kappa B mRNA expression in the colonic mucosa, as well as increase in expression of the tight junction proteins ZO-1 and occluding. Moreover, the authors should present in Introduction or Discussion that double-blind pilot trial in humans showed that oral administration of glucosamine and chondroitin modulates gut the composition of the gut microbiome which may have implications for therapeutic efficacy (PMID: 31771179).
  12. Abnormalities in colonic glycoprotein synthesis have been implicated in the pathogenesis of ulcerative colitis and Crohn's disease. Glucosamine synthetase is the rate-limiting step in the biosynthesis of gastrointestinal glycoprotein. The authors should present the relationship between clinical and mucosal glucosamine synthase activity (PMID: 852752; PMID: 856680).
  13. The authors should also mention that glucosamine and chondroitin are among the most frequently used specialty supplements, especially in patients with joint-related osteoarthritis (OA) pain (PMID: 19109115). In addition, several prospective cohort studies have shown that use of glucosamine and/or chondroitin are associated with a reduction in colorectal cancer risk (PMID: 27357024; PMID: 29411204). PMID: 33055203). Glucosamine and chondroitin supplementation has been also shown to lower systemic inflammation (PMID: 25719429).
  14. The authors should mention very interesting article by Salvatore et al.A pilot study of N-acetyl glucosamine, a nutritional substrate for glycosaminoglycan synthesis, in paediatric chronic inflammatory bowel disease (PMID: 11121904).
  15. The authors should write some sentences about the role of appropriate mucosal blood flow in the protection and healing of gastrointestinal organs. Previous studies have shown exposure of gastric mucosa to damaging factors leads to little or no damage if adequate mucosal blood flow is maintained (PMID: 7533677; PMID: 11063149). The same protective and healing-promoting effect of adequate organ blood flow has been also found in the colon (PMID: 26713317; PMID: 27433160; PMID: 30002711).
  16. References. The authors should check list of references. They should correctness of references with guide for authors. Moreover, for item 9, There is no title of the article.

Author Response

# REVIEWER 3:

The above manuscript is interesting; the authors tested the influence of N-palmitoyl-D-glucosamine, a natural amide of palmitic acid and glucosamine, given after rectal application of 2,4-dinitrobenzene sulfonic acid (DNBS) on the development of DNBS-induced colitis. However, it should be noted that the manuscript contains some deficiencies and errors that must be removed before accepting the manuscript for publication.

We thank the reviewer for their important view on our initial draft and their suggestions for improvements. We made extensive revisions to the manuscript text to address the criticisms arising from the Reviewer with a particular emphasis on editing the Material and Methods. The Reviewer highlighted many important points that needed to be better discussed and we think that this helped us to improve the content and the presentation of our manuscript. Thank you again for your help in improving our article.

1. Abstract, line 30. What does “improves colitis severity” mean? Replace “improves” with “reduces”.

Revised as suggested (P1 line 33)

2. Abstract, line 30. The authors should clearly state what kind of “alterations of intestinal barrier function” was prevented. This part of the sentence is unclear, the more so as the next sentence addresses the problem more precisely. Therefore, it would be advisable to omit this part of the sentence and also modify the beginning of the next sentence.

Revised as suggested (P1 line 34)

3. Section “2.1. Animals and experimental design”, line 108. The authors should present the total number of animals used in the study.

Revised as suggested (P3 line 121).

4. Line 110 and next lines. All abbreviations should be presented in their full name at the point where they appear for the first time. Full names of abbreviation should be repeated in the body of the manuscript at the place of the first use, as well as in tables and Figure legends. Tables and Figures should be understandable without carefully reading the text of the manuscript. On the other hand, the number of abbreviations should be reduced. The excess of abbreviations makes the manuscript difficult to understand.

We thank the reviewer for the suggestions, and we have revised this entire manuscript in response to these comments.

5. Line 113. What does “saline water 0.9%” mean? It should be replaced with 0.9% NaCl solution or simply saline.

We agree with the reviewer, and we replaced it with saline. Revised as suggested (P3 line 135)

6. Section 2.1. Animals and experimental design, lines 113-118. The text is messy. Each experimental group should be described separately. The authors should write that colitis was induced by intracolonic administration of DNBS. The method should be described in detail. Has colitis induction been performed in conscious or anesthetized animals? If animals were anesthetized, how was this done? How deep was the colon catheter inserted? Were animals fasted before? What was the reason for choosing the GA and MK886 doses used? Were the daily doses administered in one single dose or divided doses? In what form were PGA and MK886 served? What was obtained, solution or suspension? What was the vehicle for compounds used? What was the volume of doses administered? Were animals in control group and colitis group without administration of test agents receiving placebo (vehicle) by gavage and intraperitoneally in the volume and at the same time as the animals receiving PGA and MK886? Were the animals receiving PGA without MK886 administered anything intraperitoneally? All this information should be included in the manuscript.

We appreciated the reviewer’s suggestions and we revised the “Animal and experimental design” section accordingly (P3 line 121).

7. Section 2.1. Animals and experimental design, lines 120-121. The authors stated that “Mice were allowed to recover for 7 days following DNBS administration”. The animals were killed on the 7thday and the authors did not check histological and biochemical parameters of the severity of colitis. For this reason, the authors had no information that there was any recovery between the firs and the seventh day of the study.

We thank the reviewer for raising this point that needed to be better described. After colitis induction on day 0, mice were daily monitored and changes in their body weight, stool consistency, and general behavior were recorded to assess the colitis severity by the disease activity index (DAI). DNBS-treated mice lose usually around 10% of their body weight within the firsts 2-3 days and may display bloody and loose stools with hyporeactive behavior. These parameters progressively flat to those before DNBS treatment on day 5-7 post-colitis, suggesting that the acute phase of the inflammation is resolved.

8. Section 2.1. Animals and experimental design, line 123. How were animals euthanized? The authors should present the method of euthanasia in detail.

Revised as suggested (P4 line 165)

9. Lines 61-62. In the case of all drugs, regents, assays and equipment, the authors should provide general and trade name of the reagent or equipment, manufacturer’s name, city, and country.

Revised as suggested.

10. In the section 2.1. Animals and experimental design. The authors stated that they used 10 animals in each experimental group. In contrast, In Figure 1 and 3, the authors wrote that n was 5. What was the reason of these discrepancies? Was any premature animal mortality or selection of results prior to statistical analysis? This should be clearly stated in the manuscript. In the case of Figure 2 B C and D, the number of animals/observations is not known.

Thank you for bringing this to our attention. We have performed immunohistochemical analyses on the distal colon and we used N=5 colon for histopathological assessment and N=5 colon for immunofluorescence assay. For each group, colon samples were randomly split into these two analyses. We added this information to the section “Animal and experimental design” (P3 line 121).

11. The authors stated in the manuscript that “Yomogida et al. demonstrated the in-vivo suppressive effect of glucosamine on intestinal epithelial cell activation in a model of DSS-induced colitis, suggesting that the anti-inflammatory action of PGA might derive from both the amide and monosaccharide portions [11]”. The relationship between glucosamine, experimental colitis, and IBD has been studied much more widely. The observations of Yomogida et al. have been confirmed by Bak et al. (PMID: 24325781). They have found that glucosamine reduces colitis-associated symptoms in DSS-induced colitis. This effect was associated with reduction in Tumor necrosis factor-α(TNF-α), interleukin-1β, and nuclear factor-kappa B mRNA expression in the colonic mucosa, as well as increase in expression of the tight junction proteins ZO-1 and occluding. Moreover, the authors should present in Introduction or Discussion that double-blind pilot trial in humansshowed that oral administration of glucosamine and chondroitin modulates gut the composition of the gut microbiome which may have implications for therapeutic efficacy (PMID: 31771179). 

Thank you for the suggestion. We agree and that suggested references were argued in introduction and discussion sections.  

12. Abnormalities in colonic glycoprotein synthesis have been implicated in the pathogenesis of ulcerative colitis and Crohn's disease. Glucosamine synthetase is the rate-limiting step in the biosynthesis of gastrointestinal glycoprotein. The authors should present the relationship between clinical and mucosal glucosamine synthase activity (PMID: 852752; PMID: 856680).

We thank the reviewer for pointing out the importance of glutamine synthetase activity in IBD pathogenesis and we mentioned the suggested references in the Discussion section (P11 line 337)

13. The authors should also mention that glucosamine and chondroitin are among the most frequently used specialty supplements, especially in patients with joint-related osteoarthritis (OA) pain (PMID: 19109115). In addition, several prospective cohort studies have shown that use of glucosamine and/or chondroitin are associated with a reduction in colorectal cancer risk (PMID: 27357024; PMID: 29411204). PMID: 33055203). Glucosamine and chondroitin supplementation has been also shown to lower systemic inflammation (PMID: 25719429).

We thank the reviewer and added references about the glucosamine and/or chondroitin capability to reduce colon rectal cancer risk and inhibit systemic inflammation in discussion. (P11 line 351)

14. The authors should mention very interesting article by Salvatore et al. “A pilot study of N-acetyl glucosamine, a nutritional substrate for glycosaminoglycan synthesis, in paediatric chronic inflammatory bowel disease (PMID: 11121904).

We mentioned the suggested reference in the Discussion section (P11 line 351)

15. The authors should write some sentences about the role of appropriate mucosal blood flow in the protection and healing of gastrointestinal organs. Previous studies have shown exposure of gastric mucosa to damaging factors leads to little or no damage if adequate mucosal blood flow is maintained (PMID: 7533677; PMID: 11063149). The same protective and healing-promoting effect of adequate organ blood flow has been also found in the colon (PMID: 26713317; PMID: 27433160; PMID: 30002711).

We appreciate the reviewer suggestion and we recognized the importance to mention how the regulation of mucosal blood flow might encompass some aspects of PGA anti-colitis activity (P13 line 422).

16. References. The authors should check list of references. They should correctness of references with guide for authors. Moreover, for item 9, There is no title of the article.

Revised as suggested.

Round 2

Reviewer 2 Report

The paper looks generally OK.  Because of the numerous alterations, please recheck the grammar and spelling, e.g. line 432, page 10, "glucosamine synthetase.  This enzyme sintheize"

Author Response

#REVIEWER 2

The paper looks generally OK.  Because of the numerous alterations, please recheck the grammar and spelling, e.g. line 432, page 10, "glucosamine synthetase.  This enzyme sintheize"

We are pleased that our revised version has satisfied the critical issues identified by the reviewer. We thank the reviewer for help in improving the content and form of the manuscript. We have revised the sentence accordingly and rechecked the grammar and spelling of the manuscript.

Reviewer 3 Report

Manuscript # biomolecules-1853993

Title: N-palmitoyl-D-glucosamine inhibits TLR-4/NLRP3 and improves DNBS-induced colon inflammation through a PPAR-alpha-dependent mechanism

Authors: Irene Palenca et al. 

The new version of the manuscript is almost ready for publication. However, there are still some minor deficiencies presented in comment 9. In this comment, the reviewer suggested that in the case of all drugs, regents, assays and equipment, the authors should provide general and trade name of the reagent or equipment, manufacturer’s name, city, and country. The authors responded that “revised as suggested”. However, there are still some reagents with no  city name (for example line 150, 158, 159, 237, 238, 239, 278) or country name (for example line 167). The Abbreviations MA or TX are not city names, but US state codes.

Author Response

#REVIEWER 3

The new version of the manuscript is almost ready for publication. However, there are still some minor deficiencies presented in comment 9. In this comment, the reviewer suggested that in the case of all drugs, regents, assays and equipment, the authors should provide general and trade name of the reagent or equipment, manufacturer’s name, city, and country. The authors responded that “revised as suggested”. However, there are still some reagents with no city name (for example line 150, 158, 159, 237, 238, 239, 278) or country name (for example line 167). The Abbreviations MA or TX are not city names, but US state codes.

Thanks to the reviewer for catching up on these mistakes and helping us improve the quality and form of our manuscript. We have revised accordingly and entered the names of the cities for all reagents and equipment.